# Evaluation of the Effectiveness of *Helicobacter pylori* Eradication Regimens in Lithuania during the Years 2013–2020: Data from the European Registry on *Helicobacter pylori* Management (Hp-EuReg)

**DOI:** 10.3390/medicina57070642

**Published:** 2021-06-23

**Authors:** Paulius Jonaitis, Juozas Kupcinskas, Olga P. Nyssen, Ignasi Puig, Javier P. Gisbert, Laimas Jonaitis

**Affiliations:** 1Department of Gastroenterology, Lithuanian University of Health Sciences, 50161 Kaunas, Lithuania; pjonaitis95@gmail.com (P.J.); juozas.kupcinskas@lsmuni.lt (J.K.); 2Gastroenterology Unit, Hospital Universitario de La Princesa, Instituto de Investigación Sanitaria Princesa (IIS-IP), Universidad Autónoma de Madrid (UAM), 28006 Madrid, Spain; opn.aegredcap@aegastro.es (O.P.N.); javier.p.gisbert@gmail.com (J.P.G.); 3Centro de Investigación Biomédica en Red de Enfermedades Hepáticas y Digestivas (CIBEREHD), 28029 Madrid, Spain; 4Digestive Diseases Department, Althaia Xarxa Assistencial Universitària de Manresa, 08243 Manresa, Spain; ignasi.puig@aegastro.es; 5Department of Medicine, Facultat de Ciències de la Salut, Universitat de Vic-Universitat Central de Catalunya (UVic-UCC), 08500 Manresa, Spain

**Keywords:** effectiveness, eradication, *Helicobacter pylori*, Hp-EuReg, Lithuania

## Abstract

*Background and Objectives*: The prevalence of *H. pylori* in Eastern Europe remains quite high; however, there is insufficient data on the eradication regimens and their effectiveness. Therefore, the objective of the study was to evaluate the diagnostic methods and treatment of *H. pylori* infection as well as their adherence to Maastricht V/Florence consensus during the years 2013–2020 in Lithuania. *Materials and Methods*: Sub-study of the “European Registry on *H. pylori* Management” (Hp-EuReg), international multicenter prospective non-interventional registry of the routine clinical practice. Lithuanian data from the years 2013–2020 were analyzed for effectiveness on a modified intention-to-treat (mITT) basis. 2000 adult patients, diagnosed with *H. pylori* infection, were included. Data were compared to the European Maastricht V guidelines. *Results*: Triple-therapy was used in 90% of the cases. In 91% of the first-line prescriptions, standard triple therapy (STT) was used. The most common second-line treatment was a combination of PPI, amoxicillin and levofloxacin (PPI+A+L) (47%). The overall effectiveness in 552 cases valid for analysis was 90% by mITT. In first-line treatment, the STT effectiveness was 90% and second-line treatment with PPI+A+L achieved 92% by mITT. Increasing overall *H. pylori* eradication rates were observed: from 72% in 2013 to more than 90% in 2018–2020, as well as a shift from 7 to 10–14 days treatments duration throughout 2013–2020. *Conclusions*: In Lithuania, the prescribed eradication regimens for *H. pylori* were in accordance with the international guidelines but diagnostic methods and treatment duration only partially met Maastricht V/Florence guidelines. The eradication effectiveness was improved progressively during the years 2018–2020, reaching ≥90% cure rates.

## 1. Introduction

*Helicobacter pylori* (*H. pylori*) infection causes chronic gastritis and is the main factor in the etiopathogenesis of peptic ulcer disease and gastric cancer [1,2,3,4]. In the year 1994 *H. pylori* was classified as Class I carcinogen by World Health Organization (WHO) [5] and, in fact, still remains the only bacterium given this classification [6].

It is estimated that around 50% of the World’s populations is infected with *H. pylori* [7,8]. Based on a systematic review and meta-analysis by Hooi et al. in the year 2017, Africa has the highest prevalence of *H. pylori* (70%), while the lowest prevalence is observed in the Oceania region (20%). The prevalence of *H. pylori* in Western Europe is 34%, however there is a lack of data from Eastern Europe [9]. Recent epidemiological reviews have stated that trends of continuously decreasing prevalence of this bacterium has been reported from different areas around the globe [3,10,11,12,13]. There is a lack of epidemiological *H. pylori* studies in Lithuania—already published articles have shown the prevalence of *H. pylori* ranging from 36% in children [14] up to 70% in adults [15,16]. A most recent study has shown that over the last 25 years (1995–2020) the prevalence of *H. pylori* has decreased significantly from 52% to 14% among the students of Lithuanian University of Health Sciences [17]. Moreover, the reinfection rate of *H. pylori* in Lithuania is still relatively high [18].

The diagnostics of *H. pylori* infection is divided into invasive and non-invasive methods. Invasive methods include rapid urease test (RUT), histological examination or microbiological culture. Non-invasive methods are C13 urea breath test (C13- UBT), monoclonal stool antigen test (SAT), PCR detection of *H. pylori* DNA in stool and serology [19,20,21]. The guidelines state that UBT and SAT are the recommended non-invasive tests, whereas serological tests should be only used after validation. It is recommended by the international guidelines to assess *H. pylori* eradication success after the treatment. C13-UBT and monoclonal SAT are the recommended non-invasive tests for confirmation of *H. pylori* eradication, whereas serology should not be used as a confirmatory test [1,22]. All of the invasive diagnostic methods can be used as well [1,23,24].

The European *H. pylori* diagnostics and treatment recommendations are presented in the Maastricht V/Florence Consensus report, which are also unofficially recommended to adhere in Lithuania [1]. The suitable *H. pylori* treatment should be selected after the evaluation of previous eradication regimens (first-line, second-line etc.) and the antibiotic resistance of the bacterium in the selected region. Standard triple therapy (PPI, clarithromycin and amoxicillin) is the recommended first-line eradication regimen in regions with low (<15%) clarithromycin resistance. Bismuth based quadruple therapy or non-bismuth concomitant treatment is the recommended first-line treatment in areas with high clarithromycin resistance (>15%). If the first-line eradication therapy is not successful, bismuth based quadruple therapy or levofloxacin-based therapy is recommended by the guidelines [1,25,26]. The recommended treatment duration is 14 days and 10 days therapies should only be used when proven effective locally.

*H. pylori* resistance rates to antibiotics are increasing worldwide. An extensive analysis by Megraud et al. in the year 2019 reported that primary clarithromycin resistance in European countries has doubled in the last 20 years [27]. According to the study the mean clarithromycin resistance in Europe is 21.6%, however Lithuania still belongs to the area with low (<15%) clarithromycin resistance with estimated resistance of 13% [27]. Studies in Lithuania have revealed that *H. pylori* resistance to clarithromycin was 8.2% in adults and 17.7% in children during the years 2013–2015 [28].

According to recommendations, the optimal *H. pylori* eradication rate is ≥90% [29,30,31]. The data from the European registry on *H. pylori* management (Hp-EuReg) has shown a decline in the use of triple therapy from over 50% prescriptions in the years 2013–2015 to less than 32% in the years 2017–2018. A trend of an increasing use of quadruple therapy was observed during the years 2013–2018. The same study concluded that standard triple therapy remains the most popular first-line eradication regimen and an increase in first-line *H. pylori* eradication success from 74% to 88% was observed. The overall *H. pylori* eradication rate in Europe has increased from 83.9% to 87.8% during the years 2013–2018 [32]. However, some studies have shown a worrying decline in the eradication rate of standard triple therapy due to the increasing antibiotics resistance [33,34]. It has been calculated that the eradication rate of standard triple therapy is 84% after 10 days of treatment and 86% after 14 days of treatment. The optimal eradication rate (>90%) was only achieved in various quadruple regimens, including bismuth based schemes, after at least 10 days of treatment [32]. Therefore, the previously used 7 days treatment duration is no longer recommended and the optimal *H. pylori* eradication rate can be only achieved with 10–14 days treatment [35,36,37].

Therefore, the aim of this sub-study was to evaluate the *H. pylori* diagnostics, treatment prescriptions, their effectiveness and adherence to Maastricht V/Florence consensus report during the years 2013–2020 in Lithuania.

## 2. Materials and Methods

### 2.1. European Registry on H. pylori Management (Hp-EuReg) Setting and Ethics

The Hp-EuReg is an international, multicenter, prospective, non-interventional registry that has been recording information on the management of *H. pylori* infection since 2013. The Hp-EuReg protocol [38] establishes national coordinators in the currently selected 31 countries, where gastroenterologists have been recruited at over 300 study centers to provide input to the registry. Lithuania is one of the participating countries.

This study was performed at the Department of Gastroenterology of Lithuanian University of Health Sciences (LUHS). It is the main Hp-EuReg center in Lithuania and collects data from other hospitals as well. The study was approved by the Ethics Committee of La Princesa University Hospital, Madrid, Spain (20/12/2012, Nr. v.04-12-12 and 15/01/2015, Nr. v.22-12-14) and by the Bioethics Center of LUHS (19/11/2018, Nr. BEC-MF-94). It was prospectively registered at ClinicalTrials.gov (NCT02328131) (accessed on 1 February 2021).

### 2.2. Participants

The data of the study has been obtained from Lithuanian patients that were included in the Hp-EuReg during the years 2013–2020. All of the patients were adults, who were diagnosed with *H. pylori* infection.

### 2.3. Data Extraction and Analysis

Data were collected through an electronic Case Report Form (e-CRF), collecting the patient’s demographic information, any previous eradication attempts and the treatments employed, the outcomes of any treatment, recording details such as the compliance, the cure rate, the follow-up, etc. and any adverse event (AE). This information was registered at the REDCap database [39] managed and hosted by the “Asociación Española de Gastroenterología”, Madrid, Spain (AEG; www.aegastro.es; accessed on 1 January 2021), a non-profit Scientific and Medical Society that focuses on Gastroenterology research.

### 2.4. Effectiveness Analysis

In order to calculate the effectiveness of different *H. pylori* eradication regimens, a modified intention-to-treat (mITT) analysis was used in order to reach the closest result to those obtained in clinical practice. The mITT includes all of the cases that had completed the follow-up (i.e., had undertaken a valid confirmatory test after the eradication treatment), regardless of the compliance to treatment and excluding those cases with an incomplete follow-up (lost to follow-up). All the patients, that were empirically treated, were included in the effectiveness analysis, excluding the cases, in which serology was used as a method of confirmation after the eradication treatment because it is not recommended in the guidelines [1].

### 2.5. Statistical Analysis

Continuous variables were summarized as the mean and standard deviation, while qualitative variables were presented as the absolute relative frequencies, displayed as percentages (%). Qualitative variables are presented as percentages and 95% confidence intervals (95% CI), where applicable. Statistical analysis was performed using IBM SPSS Statistics ( Version 25.0, Armonk, NY, USA) and Microsoft Office Excel 365 (Redmond, WA, USA). The selected level of statistical significance was *p* < 0.05.

## 3. Results

Figure 1 depicts the flow-chart of the study. During the years 2013–2020, data from 2000 cases were included in the Hp-EuReg in Lithuania and used for further analysis. The mean (SD) age of the patients was 50 (15.5) years. There were 1137 female (57%) and 862 male (43%) patients. The main indications for the treatment of *H. pylori* infections were functional dyspepsia (20%), duodenal (13%) and gastric ulcers (8%). Other causes (chronic gastritis, gastroesophageal reflux disease, erosive gastritis etc.) accounted for 53% of the indications but were not registered separately in the registry. The main *H. pylori* diagnostic methods before the eradication were rapid urease test (996 cases; 50%), serology (665 cases; 33%), histological examination (334 cases; 17%), stool antigen test (44 cases; 2%) and urea breath test (27 cases; 1%).

The diagnostic methods used to evaluate eradication success after the treatment were also analyzed. The most frequent diagnostic methods were rapid urease test (227 cases; 41%), histological examination (216 cases; 39%), stool antigen test (145 cases; 26%), microbiological culture (2 cases; 0.4%) and urea breath test (11 cases; 2%). Serology was used as a diagnostic method after the treatment in 16% of the cases, however, these records were excluded from further mITT analysis.

The main *H. pylori* eradication regimens were as follows: triple therapy (PPI + 2 antibiotics) in 1796 cases (90%), bismuth or non-bismuth based quadruple therapy (PPI + bismuth + 2 antibiotics or PPI + 3 antibiotics (clarithromycin, amoxicillin, metronidazole)) in 194 cases (10%), sequential therapy in 5 cases (0.3%) and dual therapy in 3 cases (0.2%).

The most frequently prescribed PPIs were also analyzed: the most popular were omeprazole (43%), esomeprazole (34%), pantoprazole (12%) and rabeprazole (9%). In 92% of the cases the PPIs were given twice a day and in 8% of the cases they were given once a day.

The most prescribed *H. pylori* eradication scheme was standard triple therapy, which consists of a PPI, clarithromycin and amoxicillin—in 79% of the cases. The second most prescribed regimen was a triple therapy, consisting of a PPI, amoxicillin and levofloxacin—it was prescribed in 8% of the overall cases and was usually used as the main second-line prescription. Bismuth and non-bismuth-based quadruple therapies were prescribed quite rarely; however, they have been used more frequently during the last three years. Table 1 shows the prescription frequency of various first and second-line regimens, the number of patients tested after the treatment and the mITT effectiveness of each eradication scheme.

The effectiveness of various eradication therapies was calculated after analyzing 552 completed cases, which represented only 29% of the total cases after serology (*n* = 77) exclusion. Significant amount (*n* = 1371; 71%) of the remaining 1923 cases were incomplete or lost to follow-up.

Among 552 cases evaluated after the treatment, the overall effectiveness of *H. pylori* eradication was 90% (95% CI: 87–92%). Regarding the trends of *H. pylori* eradication success during the years 2013–2020, an increasing eradication rate from 72% in the year 2013 to more than 90% during the years 2018–2020 (94%, 97.8% and 92.6% respectively) has been observed (Figure 2).

The overall effectiveness of first-line standard triple therapy was 90% (95% CI: 86–92%) and was 92% (95% CI: 80–98%) of the main second-line therapy, consisting of PPI, amoxicillin and levofloxacin. The effectiveness of most other first and second-line eradication regimens was 100% but these data are not reliable due to the small sample size of treated patients.

Of all 2000 cases, the average (SD) duration of the *H. pylori* infection treatment was 10 (2.7) days. The eradication therapy was prescribed for 7, 10 or 14 days and the distribution of treatment duration in first and second-line treatments is presented in Figure 3. The most often prescribed duration was 7 days in more than 40% of first-line therapy patients and was 10 days in more than a half of second line treatment cases. The most frequent duration of *H. pylori* eradication prescriptions in the years 2013–2017 was 7 days, however, starting from the year 2015 a clear shift to 10 or 14 days treatment durations was observed and from the year 2018 10–14 days treatment durations became the most frequently prescribed. The visual analysis of these changes is presented in Figure 4.

## 4. Discussion

In this study it was evaluated if the Hp-EuReg data from Lithuania during 2013–2020 met the guidelines of Maastricht V/Florence consensus report.

A worrisome fact that the confirmatory tests after *H. pylori* eradication were not a part of a standard routine clinical practice was revealed by our analysis and 1371 cases (71% of the cases) were incomplete or lost to follow-up, allowing only 552 patients (29%), who were tested after the treatment, to be valid for the effectiveness analysis. It could be speculated that some of these patients may have undergone a control diagnostic test after the eradication in other health care institutions and did not provide the information about that. However, it is most likely that these patients did not undergo confirmatory tests as they were not sufficiently motivated to do it by the responsible healthcare professional or by themselves. As the prevalence of *H. pylori* in Lithuania is still quite high [14,15,16,17], a much higher rate of treatment evaluation could result in better treatment outcomes and more rapid decrease in the prevalence of this infection. We would also like to state that the only way to improve *H. pylori* cure rates is to better know what the current situation of *H. pylori* diagnostics and treatment in Lithuania looks like, and for doing that, confirmation of *H. pylori* eradication as a general rule is necessarily needed.

On the other hand, an obvious lack of valid non-invasive tests in Lithuania was observed. Stool antigen tests were introduced to routine clinical practice in the year 2018 and C^13^-UBT tests have only been available since the end of 2019. There are just a few health care institutions available to perform C^13^-UBT—mainly in larger cities and it is not reimbursed by the state. Therefore, it is not surprising that a lot of the patients did not undergo confirmatory tests or the serological assay has been used as a confirmatory test, as it is much more convenient and cheaper. However, as serology is not recommended as a confirmatory test post treatment, many cases could not be included in the effectiveness analysis. There is a lack of published data but it is most likely that similar situation can be observed in many Central-Eastern European countries, especially in smaller cities and rural areas. Therefore, our findings may serve as a challenge to gastroenterologists and general practitioners to raise this question to local health care authorities.

As already mentioned, the UBT is the main recommended non-invasive diagnostic test for the primary diagnosis of *H. pylori* infection, as well as the main non-invasive diagnostic option for confirmation of *H. pylori* eradication after the treatment. Monoclonal SAT is an alternative in both scenarios. Only validated serologic tests can be used for the primary diagnostics of *H. pylori* and are not recommended for the evaluation of treatment success. It was clearly shown by our research that serology was one of the main non-invasive diagnostic options not only for the primary assessment of *H. pylori* infection but, despite the recommendations, it was also used for the confirmation of *H. pylori* eradication. It is not the best recommended non-invasive method to use for the primary diagnostics of *H. pylori* and should not be used for the treatment evaluation. It is expected that the availability and use of UBT and SAT should increase in the upcoming years. Rapid urease test was the main method for the invasive diagnostics of *H. pylori* and meets the current recommendations as a first-line diagnostic test.

Concerning *H. pylori* eradication regimens, the use of the main first and second-line prescriptions in Lithuania during the years 2013–2020 met the Maastricht V/Florence consensus report. The most prescribed first-line therapy was a standard triple therapy (PPI, clarithromycin, amoxicillin—91% of the cases) and Lithuania still belongs to the area of low clarithromycin resistance (<15%) [27,28]. Levofloxacin containing triple therapy is one of the recommended second-line treatment regimens and, in fact, was the most widely used (47% of the second-line cases) scheme for the second line treatment in Lithuania. It was observed in our sub-study that the use of various bismuth and non-bismuth-based quadruple therapies has been increasing as well and is expected to become even higher in the future. As recommended by the guidelines, the PPIs were mostly administered twice daily but there were some cases (8%) when they were given only once a day and this should be avoided as most of the studies have clearly shown that double doses of PPIs were superior to a single dose for the effectiveness of *H. pylori* eradication regimens [40,41]. The guidelines state that the recommended treatment duration of *H. pylori* infection should be 10 to 14 days. During the years 2013–2020 in Lithuania the most frequent duration of *H. pylori* eradication therapy was 7 days for the first-line and 10 days for the second-line treatment. It should be noted that even though the 7 days treatment duration is no longer recommended by international guidelines, it was still the most frequent during the years 2013–2020 in Lithuania. However, an obvious trend of increasing 10 to 14 days treatment durations from the year 2016 was observed and 14 days treatment duration even became the most frequently prescribed in the year 2020. This shift from 7 to 10–14 days treatment duration was associated with the acceptance of updated Maastricht V/Florence consensus report in the year 2016, where 7 days duration was no longer recommended.

One of the main goals of this research was to evaluate the effectiveness of various *H. pylori* eradication regimens. During the years 2013–2020 the effectiveness of first-line standard triple therapy was 90% and of second-line PPI, amoxicillin, levofloxacin was 92%. The effectiveness of the main first-line and the main second-line prescription was optimal (≥90%). The overall *H. pylori* eradication effectiveness during the years 2013–2020 was optimal (90%) as well and the eradication effectiveness has even reached >90% success rates during 2018–2020 and met the recommended goal of Maastricht V/Florence consensus report.

Similar studies have been performed in other Central and Eastern European countries and the results are diverse. The results in Kazan (Russia) have also shown a low follow-up after the eradication treatment as well as suboptimal (<90%) treatment effectiveness even in 14 days duration standard triple therapy [42]. Another Russian study concluded that the success rate of the most popular 10 days duration standard triple therapy was only 79% during the years 2013–2018 and serological test were still used for the evaluation of treatment effectiveness [43]. A study in Hungary has shown a suboptimal overall eradication success rate of the first line therapy during the years 2013–2019 and only 10-day quadruple concomitant treatment regimen achieved 95.1% effectiveness. What is more, second- and third-line eradication regimens achieved largely suboptimal results (65.2 and 54.5% respectively) [44]. A Slovenian study from the years 2013–2015 has concluded that even though only 11.4% of the total 1519 cases were lost to follow-up, the mITT eradication success rates were also suboptimal: the overall treatment effectiveness of standard triple therapy was only 72% and of the main second-line PPI, amoxicillin and levofloxacin regimen was 87.1% [45]. However, a more recent Slovenian study from the years 2017–2019 has clearly shown improvements: the 14 days duration first-line standard triple therapy has achieved optimal 93% success rate and the effectiveness of the main second-line 14 days regimen was 89% [46]. The already mentioned data from 21,533 cases from Hp-EuReg has shown significantly increasing *H. pylori* eradication rates in Eastern Europe (from ~62% to more than 80%) and eradication rates ranging from ~80% to >90% in other parts of the Europe [32].

Finally, the strengths and limitations of our study should be noted. There is a clear lack of data on *H. pylori* diagnostics and treatment not only in Lithuania but in surrounding countries as well, therefore our research contributes to additional data from the region and points out the most important *H. pylori* treatment information. In addition, even though it was clearly shown by our research that the diagnostics (e.g., the use of serology for the evaluation of eradication success; extremely low use of C13-UBT) and treatment (e.g., suboptimal eradication rates) of *H. pylori* only partially meet the Maastricht V/Florence consensus report, these findings should encourage the physicians to better adhere to the guidelines and, as a result, contribute to better diagnostics and treatment outcomes of this bacterial infection. On the other hand, the main limitations of our sub-study include the lack of available proper diagnostic methods in Lithuania (such as C13-UBT devices), the high use of serology not only for the primary diagnostics of *H. pylori* but also for the evaluation of treatment effectiveness and the extremely low follow-up after prescribing eradication treatment (only 29%), which resulted in the fact that only the main first and second-line prescriptions could be properly evaluated.

## 5. Conclusions

After evaluating the diagnostics of *H. pylori* infection, treatment prescriptions and their effectiveness in Lithuania during the years 2013–2020 it was concluded that the diagnostic and treatment duration of *H. pylori* infection only partially meet the Maastricht V/Florence guidelines. The prescribed eradication regimens are in accordance with the international guidelines. The eradication effectiveness has been improved significantly during the years 2018–2020, reaching ≥90% cure rates.

## Figures and Tables

**Figure 1 medicina-57-00642-f001:**
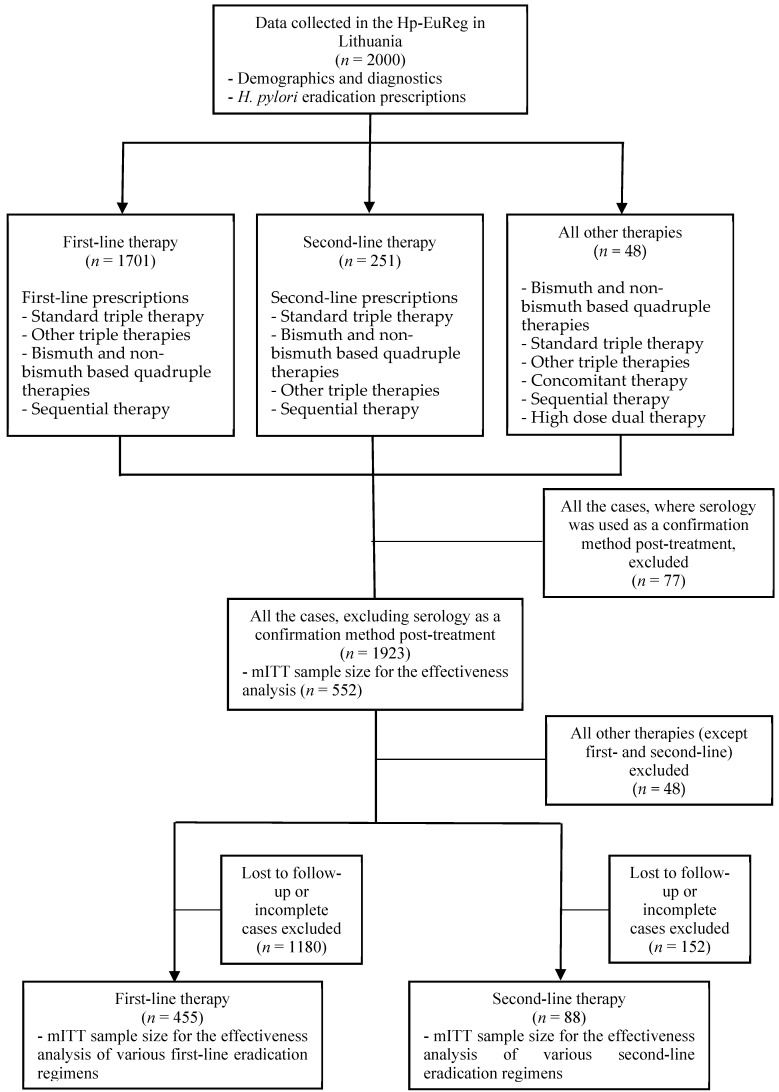
Study flow chart. mITT—modified Intention-to-treat.

**Figure 2 medicina-57-00642-f002:**
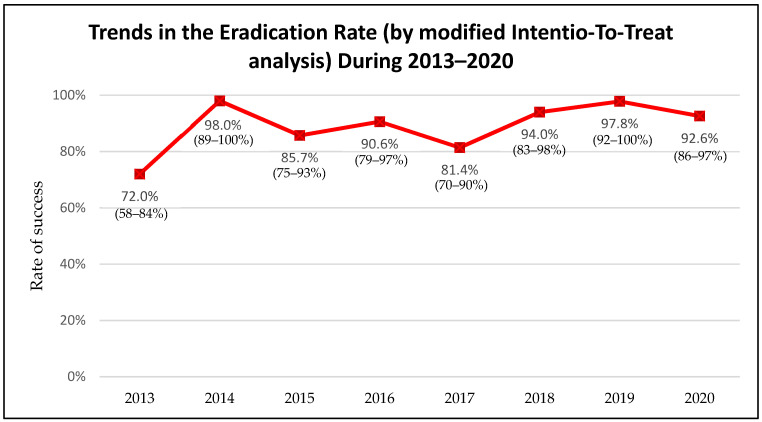
Trends in the *H. pylori* eradication rate during the years 2013–2020 (95% confidence intervals).

**Figure 3 medicina-57-00642-f003:**
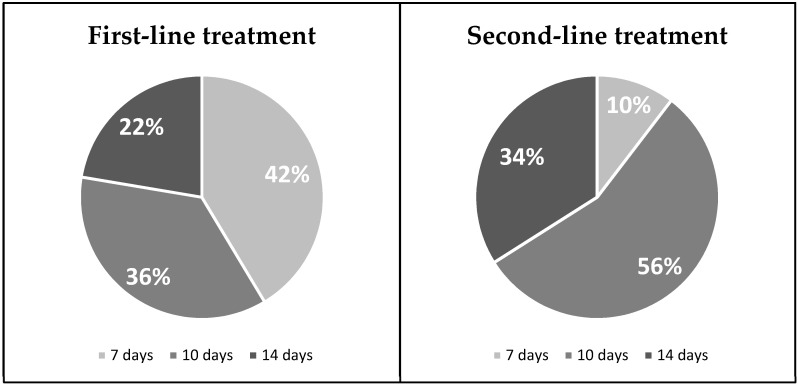
Duration of first and second-line treatments. Significant statistical differences have been found between treatment durations: χ^2^ test = 115.9; *p* < 0.001.

**Figure 4 medicina-57-00642-f004:**
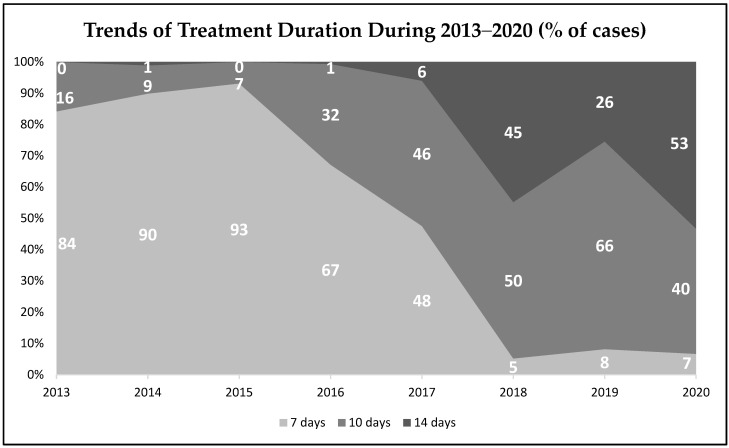
Trends of *H. pylori* eradication treatment duration during the years 2013–2020.

**Table 1 medicina-57-00642-t001:** First and second-line prescriptions and their modified intention-to treat (mITT) effectiveness. A—amoxicillin; B—bismuth; C—clarithromycin; L—levofloxacin; M—metronidazole; PPI—proton pump inhibitor; T—tinidazole; 95% CI—95% confidence interval.

**First-Line Treatment**	**No. of Patients** **(% of All Cases) **	**No. of Patients** **Tested Post-Treatment**	**mITT (95% CI)**
PPI+C+A	1550 (91%)	416	90% (86–92%)
PPI+C+A+B	66 (3.9%)	9	89% (52–100%)
PPI+C+M	34 (2%)	12	100% (74–100%)
PPI+A+L	16 (0.9%)	6	100% (54–100%)
PPI+A+M+B	10 (0.6%)	3	100% (29–100%)
PPI+A+L+B	8 (0.5%)	1	100% (3–100%)
PPI+A+M	8 (0.5%)	6	100% (54–100%)
PPI+C+L	2 (0.1%)	-	-
PPI+A+C+T (Sequential)	1 (0.1%)	1	100% (3–100%)
PPI+C+M+B	1 (0.1%)	-	-
PPI+C+A+M	1 (0.1%)	1	100% (3–100%)
**Second-Line Treatment**	** No. of Patients** **(% of All Cases) **	** No. of Patients** **Tested Post-Treatment**	** mITT ** ** (95% CI) **
PPI+A+L	118 (47.0%)	49	92% (80–98%)
PPI+A+L+B	58 (23.1%)	8	100% (63–100%)
PPI+C+A	25 (10%)	14	71% (42–92%)
PPI+A+M+B	14 (5.6%)	1	100% (3–100%)
PPI+A+M	13 (5.2%)	7	71% (29–96%)
PPI+C+A+B	11 (4.4%)	3	100% (29–100%)
PPI+C+M	5 (2%)	3	100% (29–100%)
PPI+A+C+T (Sequential)	3 (1.2%)	2	100% (16–100%)
PPI+C+M+B	2 (0.8%)	1	100% (3–100%)
PPI+C+L+B	1 (0.4%)	-	-

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
