# Peer review of "Evaluation of the Effectiveness of Helicobacter pylori Eradication Regimens in Lithuania during the Years 2013–2020: Data from the European Registry on Helicobacter pylori Management (Hp-EuReg)"

_medicina, 2021, doi:10.3390/medicina57070642_

Round 1

Reviewer 1 Report

This manuscript by Paulius Jonaitis et al. describe interesting data on the efficacy of Hp eradication regimen in Lithuania, based on european cohort data.

This manuscript suffers from limitations that did not allow to be published for now.

Global : 

must be italicized: "et al."; "e.g.", "i.e."

Prefer passive form

Introduction: 

Hp diagnosis could be based also on PCR in stool. See for example Pichon et al. JCM 2020

Methods:

Used version for statistical softwares must be indicated 

How have authors considered multiple testing approaches ?

How was considered the number of patients to include in this study (as some results seem to present a lack of power)

Results : 

Give demographic comparison in regards to global Lithuanian prevalences values. For example, the M/F ratio is pretty surprising.

Flow chart is interesting but incomplete. Give detail about other therapeis that were excluded for further analysis, give the number of lost to follow-up patients.

The CI95 is lacking in the table 1 and in the Figure 2. Even in absence of a sufficient power, this value must be indicated in order to present this possible bias.

On the figure 2 present data on the different assocation proportions and therapies duration. Have the authors search for a statistical assocation? If not, please perform.

Figure 3 is of few interest, please transfer to suppl. material.

Author Response

Dear Reviewer,

We would like kindly to thank You for the revision of our manuscript and all of the comments. Below You will find our responses to each individual comment.

Point 1: Must be italicized: "et al."; "e.g.", "i.e."

Response 1: All of the Latin abbreviations have been italicized throughout the manuscript

Point 2: Prefer passive form.

Response 2: Active form sentences have been changed to passive form sentences throughout the manuscript where applicable (e.g. lines 68-73; lines 169-170; lines 191-194; lines 206-207; lines 212-215 and other).

Point 3: Introduction: HP diagnosis could be based also on PCR in stool. See for example Pichon et al. JCM 2020.

Response 3: This non-invasive diagnostic method is not available in Lithuania and is still not frequently used worldwide as there are other more convenient diagnostic approaches; however, based on the recommendation this diagnostic method has been mentioned in the introduction section.

Point 4: Methods: used version for statistical software must be indicated.

Response 4: The versions of used statistical software have been indicated. “Statistical analysis was performed using IBM SPSS Statistics 25.0 and Microsoft Office Excel 365”

Point 5: How have authors considered multiple testing approaches?

Response 5: The investigator had the opportunity to select multiple diagnostic methods; however multiple testing approaches have not been obligatory in this study due to higher financial burden as well as inconvenience for the patients to undergo several diagnostic tests. The used diagnostic methods for the initial diagnosis of H. pylori have been validated and are approved by international guidelines; therefore any of these methods were sufficient to confirm H.pylori infection; however, some of the recommended diagnostic methods (such as C13-UBT) still were not available in Lithuania during the study period; As already mentioned in the manuscript, some of the patients have undergone several diagnostic tests for the initial diagnostics of HP, as well as the confirmation of eradication after the treatment.

Point 6: How was considered the number of patients to include in this study (as some results seem to present a lack of power)?

Response 6: The number of the patients to be included in the study is not planned and there were no intentions to calculate the power of the study. Hp-EuReg study is the collection of real-life data and it includes all of the administered H. pylori eradication cases. The number of included patients is different in various countries. The latest data from HP-EuReg has been updated in 2021 and can be found in the journal “Gut” (reference: PMID: 32958544).

Our study included 2000 patients in Lithuania because we considered it to be a sufficient number of cases to perform the analysis of H. pylori diagnostics and treatment in Lithuania and to present our data as a scientific manuscript. We are not able to properly evaluate some of the treatment regimens due to insufficient number of cases; however this issue has been addressed in the manuscript.

Point 7: Give demographic comparison in regard to global Lithuanian prevalence values. For example, the M/F ratio is pretty surprising.

Response 7: The male (43%) to female (57%) ratio indicates the situation in Lithuania – more female patients refer to the doctors due to various health issues. We could not figure out which of these situations were surprising for the reviewer – is this ratio too low or, on the contrary, this ratio is not that far to being 1. In our opinion this ratio is quite favourable because in other Lithuanian studies the M/F ratio were even lower. However, based on other studies, this ratio has no connection to the prevalence of HP – one of the Lithuanian studies found no significant difference in seroprevalence of HP between the genders in the years 1995, 2012, 2016 and 2020 [reference: PMID: 33803389].

Point 8: Flow chart is interesting but incomplete. Give detail about other therapies that were excluded for further analysis, give the number of lost to follow-up patients.

Response 8: The number of lost to follow-up patients has been added to the flow chart. Most of the excluded therapies (third-line and other) were bismuth and non-bismuth based quadruple therapies which were not further analysed during the study, therefore not relevant in the current study. In our opinion, adding additional information to the flow chart would make it more complex (and larger in dimensions) and, as a result, more difficult to read.

Point 9: The CI95 is lacking in the table 1 and in the Figure 2. Even in absence of a sufficient power, this value must be indicated in order to present this possible bias.

Response 9: The 95% confidence intervals have been added to Table 1 and Figure 2.

Point 10: On the Figure 2 present data on the different association proportions and therapies duration. Have the authors search for a statistical association? If not, please perform.

Response 10: We assume this comment was addressed to Figure 3 as Figure 2 displays the trends of eradication rates during 2013-2020. Concerning the statistical association - significant statistical differences have been found between treatment durations and this has been added in the footnote of Figure 3.

Point 11: Figure 3 is of few interests, please transfer to suppl. material.

Response 11: In our opinion Figure 3 clearly represents one of the main H. pylori eradication problems of the first-line treatment – 7 days duration treatment regimens were the most popular during the years 2013-2020. It also shows that the duration of second-line treatments were mostly 10-14 days and correspond to the treatment guidelines. If the reviewer and the editorial team agrees, we would like to keep Figure 3 in the manuscript as it is.

We hope that our changes and comments are suffice and the manuscript can be published. Thank You once again for Your time.

Regards,
Authors

Reviewer 2 Report

I appreciate the opportunity to review this interesting report on the effectiveness of Helicobacter pylori eradication regimens in Lithuania during the years 2013-2020.

In their paper, the authors have evaluated the H. pylori diagnostics, treatment prescriptions, their effectiveness, and adherence to Maastricht V/Florence consensus report during that time. They have found that the diagnostic and treatment duration of H. pylori infection only partially meets the Maastricht 302 V/Florence guidelines. The prescribed eradication regimens are following the international guidelines. The eradication effectiveness has been improved significantly during the years 2018-2020.

I commend the authors for many strengths of their work, including addressing an interesting and timely question and well-performed analysis.

The subject is in the range of the journal, and the manuscript is of clinical relevance. It is well written, and the data are appropriately presented.

However, I believe that the work would benefit if the authors showed the problem in a broader perspective and compared in the Discussion, the results of their work with the situation in other countries, particularly of Central and Eastern Europe.

Author Response

Dear Reviewer,

We would like kindly to thank You for the revision of our manuscript and all of the comments. Below You will find our responses to Your comment.

Point 1: I believe that the work would benefit if the authors showed the problem in a broader perspective and compared in the Discussion, the results of their work with the situation in other countries, particularly of Central and Eastern Europe.

Response 1: Some of the current situation (H. pylori treatment prescriptions and their effectiveness) in Europe has been described in the introductions section (lines 83-98). The latest data from HP-EuReg were published in the journal “Gut” and has been updated in the year 2021 [reference: PMID: 32958544]. In that manuscript the results and data from separate European regions are presented separately and it is possible to see the differences. There are little published data from separate countries; however, we added a paragraph in the Discussion section (lines 285-305) which describes the situation in other Eastern and Central European countries, including Russia, Slovenia and Hungary.

“Similar studies have been performed in other Central and Eastern European countries and the results are diverse. The results in Kazan (Russia) have also shown a low follow-up after the eradication treatment as well as suboptimal (<90%) treatment effectiveness even in 14 days duration standard triple therapy [42]. Another Russian study concluded that the success rate of the most popular 10 days duration standard triple therapy was only 79% during the years 2013-2018 and serological test were still used for the evaluation of treatment effectiveness [43]. A study in Hungary has shown a suboptimal overall eradication success rate of the first line therapy during the years 2013-2019 and only 10-day quadruple concomitant treatment regimen achieved 95.1% effectiveness. What is more, second- and third-line eradication regimens achieved largely suboptimal results (65.2 and 54.5% respectively) [44]. A Slovenian study from the years 2013-2015 has concluded that even though only 11.4% of the total 1519 cases were lost to follow-up, the mITT eradication success rates were also suboptimal: the overall treatment effectiveness of standard triple therapy was only 72% and of the main second-line PPI, amoxicillin and levofloxacin regimen was 87.1% [45]. However, a more recent Slovenian study from the years 2017-2019 has clearly shown improvements: the 14 days duration first-line standard triple therapy has achieved optimal 93% success rate and the effectiveness of the main second-line 14 days regimen was 89% [46]. The already mentioned data from 21,533 cases from Hp-EuReg has shown significantly increasing H. pylori eradication rates in Eastern Europe (from ~62% to more than 80%) and eradication rates ranging from ~80% to >90% in other parts of the Europe [32].”

We hope that our changes and comments are suffice and the manuscript can be published. Thank You once again for Your time.

Regards,
Authors

Round 2

Reviewer 1 Report

This manuscript has been extensively revised by the authors. Minor modification remains to be applied.

Point 5: How have authors considered multiple testing approaches?

--> The question is focused on a statistical point of view, as multiple statistical test have been applied to the same data set. Have the authors considered a control for the inflation of the alpha risk for example ?

Point 8: Flow chart is interesting but incomplete. Give detail about other therapies that were excluded for further analysis, give the number of lost to follow-up patients.

--> Even if the answer of the authors is appropriate, I maintain that the details have to be given. The flow chart is an easy-to-read figure I do not think that the suppl. information would make it ununderstandable.

Author Response

Second Response to Reviewer 1 Comments

Dear Reviewer,

We would kindly like to thank You for the second revision of our revised manuscript and all the comments. We would also like to thank You for the suggestions during the first revision in order to improve our manuscript. Below You will find our responses to each individual comment of Your second revision

Point 1: How have authors considered multiple testing approaches?
The question is focused on a statistical point of view, as multiple statistical tests have been applied to the same data set. Have the authors considered a control for the inflation of the alpha risk for example?

Response 1: We have not performed multiple testing (meaning multiple comparisons) in our study. We did perform comparisons for the different treatment schemes using the Pearson Chi-square test which essentially reports whether the results of the cross table are statistically significant; in sum - whether the different treatment schemes evaluated were unrelated (or not) of one another. This test did not inform about which treatment was different from one other specifically but reported differences between all of them. By using such a test there was no need for a Bonferroni adjustment as already mentioned in Your revision

Point 2: Flow chart is interesting but incomplete. Give detail about other therapies that were excluded for further analysis, give the number of lost to follow-up patients.

Even if the answer of the authors is appropriate, I maintain that the details have to be given. The flow chart is an easy-to-read figure I do not think that the suppl. information would make it not understandable.

Response 2: Summarized information about first- and second-line prescriptions and other excluded therapies was added to the study flow chart.

We hope that our changes and comments are suffice and the manuscript can now be published. Thank You once again for Your time.

Regards,
Authors
